# Semi-Automated Data Labeling for Activity Recognition in Pervasive Healthcare

**DOI:** 10.3390/s19143035

**Published:** 2019-07-10

**Authors:** Dagoberto Cruz-Sandoval, Jessica Beltran-Marquez, Matias Garcia-Constantino, Luis A. Gonzalez-Jasso, Jesus Favela, Irvin Hussein Lopez-Nava, Ian Cleland, Andrew Ennis, Netzahualcoyotl Hernandez-Cruz, Joseph Rafferty, Jonathan Synnott, Chris Nugent

**Affiliations:** 1CICESE (Centro de Investigacion Cientifica y de Investigacion Superior de Ensenada), Ensenada 22860, Mexico; 2IPN (Instituto Politecnico Nacional), Tijuana 22435, Mexico; 3CONACYT (Consejo Nacional de Ciencia y Tecnología), Ciudad de Mexico 03940, Mexico; 4School of Computing, Ulster University, Jordanstown BT37 0QB, UK; 5INIFAP (Instituto Nacional de Investigaciones Forestales, Agricolas y Pecuarias), Aguascalientes 20660, Mexico

**Keywords:** data labeling, gesture recognition, environmental sound recognition, activity recognition, pervasive healthcare

## Abstract

Activity recognition, a key component in pervasive healthcare monitoring, relies on classification algorithms that require labeled data of individuals performing the activity of interest to train accurate models. Labeling data can be performed in a lab setting where an individual enacts the activity under controlled conditions. The ubiquity of mobile and wearable sensors allows the collection of large datasets from individuals performing activities in naturalistic conditions. Gathering accurate data labels for activity recognition is typically an expensive and time-consuming process. In this paper we present two novel approaches for semi-automated online data labeling performed by the individual executing the activity of interest. The approaches have been designed to address two of the limitations of self-annotation: (i) The burden on the user performing and annotating the activity, and (ii) the lack of accuracy due to the user labeling the data minutes or hours after the completion of an activity. The first approach is based on the recognition of subtle finger gestures performed in response to a data-labeling query. The second approach focuses on labeling activities that have an auditory manifestation and uses a classifier to have an initial estimation of the activity, and a conversational agent to ask the participant for clarification or for additional data. Both approaches are described, evaluated in controlled experiments to assess their feasibility and their advantages and limitations are discussed. Results show that while both studies have limitations, they achieve 80% to 90% precision.

## 1. Introduction

Data labeling, or data annotation, for pervasive healthcare, refers to the process of segmenting and assigning labels to data (usually data gathered from sensors), related to a person’s behavior, activity, mood or health state, at different timestamps [1]. The data annotation process depends on the type of data being collected (images, audio, text, etc.), as well as the conditions in which the data collection is performed.

From wellness applications that rely on fitness trackers to motivate users to adopt healthy habits, to security systems that detect unusual behavioral patterns in public places, ubiquitous computing systems increasingly rely on the inference of the activity and behavior of individuals to proactively assist users in their lifestyle [2], to continuously monitor the health of patients [3], or even to estimate symptoms of certain diseases [4].

Activity and behavior recognition is mostly performed using supervised learning algorithms, such as neural networks or decision trees [5]. The training of these supervised algorithms is performed with labeled data, or “ground truth”. Incomplete or inaccurate labeling can result in classification errors that lead to unreliable systems. Obtaining high-quality labels is thus a critical component of ubiquitous computing.

Data labeling is a demanding task that is normally performed either through direct observation of the individual performing the activity or via self-report. While the former approach is costly and impractical for large populations, the latter is error-prone [6]. Data labeling using direct observation is often performed in a lab, where individuals enact an activity, such as walking or sleeping, under controlled conditions. These conditions can affect how the activity is performed and the approach also has low ecological validity [7]. The use of self-report is convenient to label datasets from large numbers of individuals performing the activity in naturalistic conditions. The limitations of self-report, however, have been well documented and include poor recollection and social-desirability biases. This problem is exemplified by recent criticism on the use of self-report dietary data in nutrition research and public policy [8].

The ubiquity of mobile and wearable sensors is allowing researchers to collect large datasets from individuals performing activities in naturalistic conditions to train activity classification models. Labeling this data, however, is particularly challenging and expensive. It requires self-labeling approaches that reduce the burden on the user and response bias.

In this paper we propose a framework to classify approaches to data labeling. We show that with adequate tools, there are numerous advantages for self, naturalistic, online data labeling, particularly for gathering large datasets. In that vein, we present an extension of two previous early works on semi-automatic data labeling that rely on self-annotation. The first work is based on the detection of subtle hand gestures when the user is queried by a smartwatch. We extended the previous publication [9] by conducting a study in naturalistic conditions and collecting data from five additional subjects. The second work [10] is used to label home activities that have an audible manifestation. The previous publication was a proof of concept while, in this work, the proposed system, called the Intelligent System for Sound Annotation (ISSA), has been fully developed and naturalistic setting evaluations have been added.

We describe both approaches in detail and report on their accuracy and application. By using machine learning, the solutions proposed reduce the burden on the annotator, while retaining the advantages of online labeling, such as its suitability for large-scale data annotation.

The remainder of the paper is organized as follows: Section 2 presents the related work in terms of approaches to data labeling and considering different types of criteria (temporal, annotator, scenario and annotation mechanisms). Section 3 describes and presents the results of the first labeling study presented, which is about the use of gesture recognition with smartwatches. Section 4 describes and presents the results of the second labeling study presented, which is about the use of smart microphones to label audible home activities. Finally, conclusions and future directions are presented in Section 5 and Section 6, respectively.

## 2. Approaches to Data Labeling

Data labeling can be performed in numerous ways, depending on the type of data being labeled and the conditions in which the activity is being performed and observed. We classify the approaches to data annotation with respect to four different criteria (when, who, where and how): (i) Temporal (when), if the labeling is conducted during or after the activity is performed; (ii) annotator (who), that is, if the labeling is done by the individual performing the activity or by an observer; (iii) scenario (where), depending if the activity being labeled can be conducted in a laboratory under controlled conditions, or in its natural setting, in a naturalistic study; (iv) annotation mechanism (how), the annotation is conducted manually, semi-automated or fully automated. The implications of these different types of annotation criteria are elaborated in the following subsections.

### 2.1. Temporal (When: Online vs. Offline)

Data can be labeled when the activity of interest is being performed (online) or it could be labeled afterwards (offline), either by the individual being monitored or by an observer. Labeling while the activity is being performed is difficult when the activities have fine granularity or when high accuracy is required in the timing of the activity. For instance, an activity classifier using accelerometer data would require precise timestamping of when the individual started walking, or even when each step was performed. The approaches described in [11,12] are examples of this criteria. This could be better estimated by looking at a video of the activity, with the ability to zoom into each step, rewind and pause the video. The approaches presented in [13,14,15] describe tools for offline data annotation. Labeling changes in facial expressions to infer mood could also be easier to do by experts looking at video recordings, even comparing the opinions of multiple annotators to obtain more precise annotations. Privacy protection might also favour labeling later by the individual who performed the activity, allowing the deletion of data that is not desired to be shared.

### 2.2. Annotator (Who: Self vs. Observer)

There are some instances in which only the person performing the activity can do the labeling. For instance, actigraphy monitoring to map perceived exertion can only be assessed by the person experiencing it. On the other hand, experts might provide more accurate labels in other instances, such as when individuals might be impaired to label their own data in real-time without influencing their own activity. It is particularly difficult to accurately label sleeping data, for instance. It might also be difficult to motivate users to perform this activity, which could be demanding.

A recent study found that self-annotation may not be an effective way to capture accurate start and finish times for activities, or location associated with activity information [16]. In this study participants were asked to proactively label their activities, not in response to a query. In [12], users manually label the activities they perform in their normal routine using a mobile app. On the other hand, recruiting people to annotate large amounts of data can be costly, or the observer could not have enough context to accurately label the data. The data-labeling approach presented in [17] enables crowds of non-expert workers, volunteer and paid, to assign labels to activities performed by users in a controlled environment.

### 2.3. Scenario (Where: Lab vs. Natural Settings)

Early work on activity recognition often used data gathered under controlled conditions, typically in laboratory settings, in which, for instance, an individual wearing an accelerometer walked on a treadmill at given speeds. These settings facilitate data labeling, often using specialized equipment such as in sleep studies. However, these conditions could influence how the activity is performed or simply not match how we usually do things (i.e., not walking on even terrain). Additionally, using data collected under controlled conditions to recognize activities performed in a naturalistic setting reduces predictive accuracy by 30–50% [16]. Alternatively, studies performed in naturalistic settings complicate data gathering and data annotation. The individual could alter his/her behavior when being observed, or privacy concerns might simply impede direct observation or audio/video recordings. The Experience Sampling Method (ESM) [18] is an alternative for data labeling in these situations. Individuals are queried at random or pre-defined moments to report on their activities or feelings. The query could also be triggered by contextual conditions, for instance, asking informants how tired they are right after their smartwatch infers that they have woken up. The data-labeling approach introduced in [12] allows users to manually annotate activities performed in their normal routine using a mobile app.

### 2.4. Annotation Mechanisms (How: Manual vs. Semi-Automated vs. Automated)

Annotation mechanisms refer to tools that have been developed to assist in the time-consuming process of data annotation. These mechanisms can assist users to annotate data in three different ways: (i) Manually, (ii) semi-automated and (iii) fully automated. The criteria that should be considered when selecting an annotation mechanism to use are: (i) Reliability, (ii) cost and (iii) time. While using domain experts to annotate data typically results in more reliable annotations, it could be costly and time consuming. On the other hand, the use of fully automated annotation tools can reduce time and costs in relation to domain experts, but the resulting annotation may not be as accurate as those produced by a domain expert. Therefore, selecting the most suitable annotation mechanism will depend on the specific cases. Some annotation mechanisms in the context of gestures and audio and video data are described in this section.

Manual annotation not only requires expertise in the field but is also a time-consuming process. An example of a generic and platform-independent manual annotation tool is ANVIL [13], which is aimed at the annotation of audiovisual material containing multimodal dialogue. Gtrace [14] is another tool for manual annotation which can be used to create annotations (in the form of traces or sequences) related to how participants perceive the rise or fall of emotions of people shown on a computer screen. The work presented by [15] introduced ELAN, a manual annotation tool for multimedia and multimodal resources, such as speech, gestures and sign language, with support for peer-to-peer cooperation.

Semi-automated annotation allows the use of computational techniques and resources in conjunction with human domain expertise. The semi-automated semantic annotation tool for video presented in [19] is used to model video annotation information in the form of a hierarchy of events and sub-events. TotalRecall [20] is another tool for semi-automated annotation used to visualize, annotate and analyze large collections of audio and video by using signal-processing algorithms. The framework for audio analysis, MARSYAS [21], includes a semi-automated annotation tool which combines temporal segmentation and sound classification to support manual annotation by the user.

Automating the segmentation and labeling stages of annotation requires computational techniques to identify the segments in which the data of interest is contained [22], and a knowledge the database that contains all the possible data to be labeled. The automated annotation approach presented as a system in [23] integrates computer vision, audio processing and machine learning to produce automated annotations of tennis game videos. The work presented by [24] proposes the use of a number of automatic detection components to create layers of annotations that can be integrated in tools like ELAN, supporting the work of human annotators. The automated annotation tool presented in [12] is in the form of a mobile app aimed at annotating accelerometer data for human activity recognition. The performance of the automated annotation tool [12] was compared to manual annotation and it obtained 80–85% average precision rate.

### 2.5. Data Annotation for Pervasive Healthcare

Studies in healthcare often rely on self-report, in which a survey or questionnaire is used by a person to provide details about that person’s circumstances (activities, behaviours, emotions, etc.). When used for data labeling, this approach can be classified, according to the dimensions outlined above, as: (i) Offline (usually reflecting on activities and behaviours of the last few days or weeks), (ii) self-annotation (completed by the individual of interest), (iii) naturalistic (the individual is asked about genuine life experiences in a natural setting) and (iv) manual (with no assistance of automatic tools) (see Figure 1). While this self-report approach to labeling is relatively simple, inexpensive and quick, it has reliability issues because the individual is asked to recall past experiences, and because of response bias.

In contrast with self-report, laboratory studies use an annotator for data labeling, the annotation is performed generally on the spot (online), under controlled conditions (lab setting) and often manually (see Figure 1). As mentioned above, while controlled conditions could amount to more accurate labeling, the conditions might not reflect user experiences in a natural setting, producing a drop in prediction accuracy.

The ESM [18] triggers users to collect daily life experiences either at pre-defined or at random time intervals. The ubiquity of smartphones has made this approach more convenient and popular. It has some of the advantages of self-report (self-annotation, naturalistic), with the advantage of making the annotation online, and thus not relying on the person’s memory. A variant of ESM uses contextual information to trigger the query at a convenient time, for instance, by inferring that the person just returned home to ask him/her to rate the transportation mode or asking how tired he/she is when waking up [25]. The approaches proposed in this study can be considered instances of ESM.

Regarding when the query is triggered, ESM approaches can be classified as [26]: (i) Interval contingent (when subjects report at regular intervals); (ii) signal contingent (when subjects report when signalled); and (iii) event contingent (when the report is requested when a defined event occurs).

A final example, frequently used in psychology, corresponds to video data annotation, or video coding. This is particularly useful to record and analyze behaviors [27] or affects [28] in natural settings. This approach is time consuming and costly, often requiring expert coders, but it is also fairly reliable for coding complex behaviours that have visible manifestations. This approach would be classified as labeled by an observer, offline and often naturalistic. Regarding the use of tools, this approach can vary from mostly manual to automated, depending on the assistance that the tool provides for the task.

The two approaches described in this paper are instances of ESM and thus can be classified as self-annotation, online and naturalistic. Furthermore, they are semi-automated, thus reducing the burden for the user. In the first case, gesture recognition is used to reduce the effort to annotate the data; in the second case, audio processing is used to trigger audible queries to the subject. While the first approach can be used for interval, signal or event contingent querying, the second approach is mostly appropriate for event contingent annotation (when a relevant audio is detected). When used in event contingent mode the individual does not have to be cognitively aware of when to report, they just respond to a query.

## 3. Study 1: Data Labeling Using Gesture Recognition With Smartwatches

### 3.1. Motivation

In this initial study, we describe and evaluate a technique for activity labeling that uses subtle gestures that are simple to execute, and that can be inferred using smartwatches. The user receives a query in the smartwatch and performs a gesture associated to the label it wants to assign to the data. The feasibility of this approach relies on the correct classification of the data obtained from the inertial sensor. For this, we use a supervised classifier.

The proposed approach supports mobile-based Ecological-Momentary Assessment (EMA), a real-time measurement methodology that uses an electronic device to prompt a subject on a question of research interest. As in the case of ESM, one of the main concerns with this approach is interruption burden. However, it has been found that reducing the complexity of the query lowers burden, even if the frequency of the assessment increases [29]. The approach we propose aims at simplifying self-data labeling by using simple queries, but also allowing the user to answer with a simple hand gesture that will not even require the user to look at the smartwatch.

### 3.2. General Description

Activity recognition research is increasingly relying on data gathered from wearable devices, notably smartwatches and fitness wristbands. An advantage of these devices is that they are attached to the body of the user and are worn more frequently. Besides, they include additional sensors, such as heart-rate that can be useful to infer sleep stages or fatigue.

Smartwatches include gestures recognition software as part of their interaction design to activate some of their functionality. For instance, when the wearer raises and turns his arm towards his face, signaling that he/she wants to look at the watch, the screen turns on.

Most gesture-recognition approaches on smartphones and smartwatches make use of data gathered from accelerometer and/or gyroscope sensors. This has been used to infer activities such as food intake or opening a medication jar [30], or recognizing personalized gestures that can be used to assist individuals with visual impairment [31].

To implement a data-labeling tool based on hand gesture recognition, we first select the gestures according to the following criteria:A limited number of gestures to be recognized (between three and seven). This number could allow labeling applications to answer binary queries (Yes/No), ternary queries (Yes/No/I don’t know) as well as queries using likert-scale queries of five or seven items.Minimize the effort required by the user to perform the gestures.The user should have little difficulty learning and differentiating the gestures.Do not use gestures that are commonly used to operate a smartwatch.The gestures should be subtle and discrete. The user should be able to perform them while performing an activity, such as driving, and without interfering with it. The user should also be able to perform it while in different body postures (laying down, sitting, running, etc.) and in various social circumstances (in class, at a restaurant, walking in the street, etc.). The user should be able to make the gesture without others noticing it.

We selected the three gestures shown in Figure 2 that meet the criteria above. These gestures involve a small and quick hand movement to tap once the thumb with either the index, middle or ring finger. We included three additional gestures similar to the previous three, but with a double tap. We originally also considered two similar gestures involving the little finger, but our initial efforts showed that the signals generated by these gestures were very similar to those produced with the ring finger. These gestures fulfil the criteria stated above, but their subtlety makes it challenging to recognize them. We next describe our proposed approach to recognize these gestures.

#### 3.2.1. Sensors and Features

Data gathered from the accelerometer and gyroscope sensors are used to characterize the movement in the wrist associated to each gesture. The accelerometer measures the acceleration at which the sensor moves in three axes, while the gyroscope measures angular velocity in the same axes. The smartwatch is usually worn in the wrist of the non-dominant hand (usually the left) and thus the X-axis coincides with the direction of the arm with positive values towards the fingers, while the Z-axis is perpendicular to the screen of the device.

The acceleration signal is pre-processed to reduce the acceleration due to gravity, using a low-pass filter.

Next, we segment the signal estimating when the gesture begins and ends. Recording starts with the signal right after the user is prompted via the smartwatch. The prompting could be done through vibration of the device or with an audible signal. The individual, however, might wait a little before performing the gesture. We make an estimation of the maximum time required to complete the gesture after being prompted. For this we take the maximum time it took a group of users to perform the double-tap with the ring finger gesture and added 1.5 s to consider the time it takes users to react after perceiving being prompted. This time window guarantees that the gesture will be contained in the signal, except in cases where there is considerable delay by the user to initiate the gesture.

The next step consists of determining whether a gesture is contained in the signal. For this we use the Dynamic Time Warping (DTW) algorithm, comparing the signal from the gyroscope to a sample gesture. We use the signal from the gyroscope since we found the angular velocity to be more stable than accelerometer data. We use the contribution from the three axes (total angular velocity) to account for differences in orientation when performing the gesture [32].

Once we infer that a gesture is contained in the signal, we further segment the signal by estimating when the gesture starts and ends. The signal (again we use the magnitude of angular velocity) is divided into windows of approximately 240 ms each. The average angular velocity of each window is calculated, and a new, compressed signal is produced with each average value per segment. A low and a high threshold were empirically determined, from user data, to infer the presence of a gesture in the signal. When the magnitude exceeds the high threshold, we establish the possible presence of the gesture and a low threshold is used to establish where the signal starts and ends.

If the segment of the potential gesture is less than 200 ms, we conclude that it does not include a gesture and the user is prompted to repeat the gesture. To eliminate potential peaks in the signal produced by sporadic movements, we confirm that the start and end of the signal is no less than 220 ms away from a threshold above 0.4 in angular velocity. If this is not the case, the length of the signal is adjusted as shown in Figure 3. Similarly, if the signal lasts more than 1300 ms it will be rejected.

Features are then extracted from signals with a potential gesture. The following features are extracted from the signals of each axis of the two sensors (accelerometer and gyroscope): (i) Original signals *X*, *Y* and *Z*; (ii) a combination of two of them $XY(x^{2}+y^{2})$, $XZ(x^{2}+z^{2})$ and $YZ(y^{2}+z^{2})$; and (iii) total magnitude $XYZ(x^{2}+y^{2}+z^{2})$. We extract even distinct features from each of these signals: Average, maximum, minimum, standard deviation and the three quartiles, for a total of 98 (14×7) features per signal. Twelve additional features (two per each component of each sensor) are calculated from the area under the curve of the signals. Finally, 98 additional features are calculated from the Fourier transform of the signals by estimating the same seven measures mentioned above. This results in a total of 208 features.

#### 3.2.2. Gesture Classification

We compare two machine learning algorithms to infer the gestures using the features described above: (i) Support Vector Machines (SVM) and (ii) Sequential Minimal Optimization (SMO). We use the following parameters on each classifier. For SVM we use the two most common kernels: Linear and a radial basis function, while we tried several parameters for cost and gamma (g) as suggested in [33]. The best results were obtained with a cost of C = 4 and g = 0.0078125. For SMO we used Polykernel with the best results obtained for C = 11.

We normalized the 208 features in the range of [−1, 1] to avoid features of greater magnitude from having a larger influence in the classification.

### 3.3. Experimental Design

We conducted two experiments to evaluate the approach proposed. In the first one we estimate the feasibility of the approach and compare the performance of the two algorithms selected (SVM and SMO) using WEKA 3.9.1. The study design is within-subjects, in which participants were asked to perform all six gestures using a smartwatch in different body positions. Two Android smartwatches were used by each participant, an LG G100 and an ASUS ZenWatch 2.

The individuals were asked to perform each gesture while in three different body postures: (i) Standing with the arms facing down; (ii) standing with the arm bent and the watch facing the user; and (iii) sitting down with the arm resting on a pile of books on top of a table (to provide support).

A total of 15 participants were recruited for the study. Inclusion criteria included: Age between 10 and 60, no previous experience using a smartwatch and being right-handed. The exclusion criteria was individuals reporting known motor problems that could cause excessive movement in their arms/hands. Each participant was asked to perform 36 gestures, repeating twice each of the six gesture in the three postures. An audible signal was used to prompt the user for the gesture to be performed. A researcher initiated and controlled the intervention sending the prompts with a smartphone connected to the smartwatch via Bluetooth.

The second experiment was conducted to assess the use of the approach in an application. Five subjects participated in this study. They were asked 20 multiple-choice questions displayed on a monitor and were required to answer each question with three to five possible answers. The questions were very simple so that we would know which answer they were expected to provide, to focus on the precision of the recognition. In this experiment we used the previously trained model, but different people as participants.

#### 3.3.1. Data Collection

The dataset of the first experiment consists of a total of 540 samples. These included two instances of each of the six gestures performed by each of the 15 participants in three different postures (2×6×15×3=540). The dataset for the second experiment includes 100 samples, these are the gestures recorded by each of the five participants answering the 20 questions asked. Each sample in each dataset includes 15 signals, corresponding to the X, Y and Z-axis of the accelerometer and gyroscope recording, in addition to the ground-truth corresponding to the gesture performed.

### 3.4. Results

We estimated the precision of both classifiers to identify each gesture.

#### 3.4.1. Gesture Detection and Signal Segmentation

A total of 515 signals were estimated as containing a gesture. Approximately four samples were eliminated from the 90 gathered for each gesture. The rejection rate was not much different per gesture (SD = 0.013). Applying the DTW algorithm found that in 94.7% of the signals a gesture was present.

Figure 4 shows an example of the presence of a gesture being detected using DTW. Figure 4a shows the signal used as reference, corresponding to the magnitude of the angular velocity of an individual performing a tap with the index finger. Figure 4b shows the signal of another gesture that is accurately recognized as including a gesture using DTW. Finally, Figure 4c shows the signal produced by the circular movement of the wrist, which the algorithm correctly identifies as not having one of the gestures of interest.

#### 3.4.2. Gesture Recognition

A tenfold cross validation was performed to estimate the accuracy of the approach. For the SVM algorithm, we obtained an average precision of 81.5%, with tap with the index finger having the highest precision of 89.4%. The lowest precision corresponded to the double tap with the index finger (77.1%). Figure 5 shows the confusion matrix for these results. From the 85 gestures of a single tap with the index finger, nine were incorrectly classified (false negatives) and six were identified as this gesture when in fact they were not (false positive). In this case, false negatives are not as important as false positives, since the system can prompt the the user to repeat a gesture when it is not recognized.

The SMO classifier had similar results, with an average precision of 81.1%. Again, the best precision was obtained with a single tap with the index finger (89.4%) and the lowest with the double tap with the index finger (77.1%).

We also estimated the precision of the approach when using only the three gestures with a single tap. In this case the precision improved to 91.4% using SVM.

#### 3.4.3. Results from the Second Experiment

We recorded a total of 40 gestures (20 per pose). The precision obtained was of 82.7%. Once again, the highest precision was obtained for the index finger with a single tap (89.9%).

### 3.5. Discussion

In this section we present an approach focused on the use of gestures for data labeling. Once a system triggers a labeling query, the user responds performing one of a maximum of six possible hand gestures. This make it possible for a system to label an event, mood or behavior. Even though the gestures are brief and subtle, the results provide evidence of the feasibility of the approach, particularly when only two or three gestures are required to label the data.

By automatically recognizing the gesture, the system can reduce the burden of online, self-labeling. Furthermore, it is possible to interact with an intelligent system in other forms beside movement, for instance, through voice. In the following section we describe an approach focused on labeling auditory signals through a voice interface.

## 4. Study 2: Smart Microphones for the Labeling of Audible Home Activities

### 4.1. Motivation

Recently, there has been a growth of commercial smart voice assistants such as Amazon Echo, Google Home or Apple HomePod. These devices are composed of a combination of speakers, artificial intelligence and an array of microphones that allow the capture of high-quality multidirectional audio. It is estimated that around 47.3 million adults in the United States have access to a commercial smart voice assistant (https://voicebot.ai/), and that number is expected to increase.

Since, on one hand, the ubiquity and popularization of these devices have attracted consumers of varied age ranges, including older adults, and, on the other, these devices have different capabilities, such as the high-quality hands free audio capture and human computer interaction (through speakers and artificial intelligence), there are wide opportunities to develop pervasive health applications. For instance, [34] proposed to recognize the activities based on audio information and [35] proposed to recognize disruptive behaviors from the auditory context.

As in many Machine learning problems, valuable labeled data is useful for adequate audio classification. Currently, there is not a large database of environmental sounds, nor of sound captured from heterogeneous scenarios. For this reason, we propose and describe our approach on the use of devices with an array of microphones and artificial intelligence capabilities to annotate audible home activities in a semi-automated manner and present results from a feasibility initial evaluation. The approach is implemented as an Intelligent System for Sound Annotation (ISSA). Currently, this work considers the scenario where a single occupant carries out the activities. Similarly, only one microphone array has been used. The number of audible home activities is limited but relevant in terms of: (i) Their common occurrence in the daily life of a person, and (ii) their use in the area of activity recognition. A sample scenario is described in [10].

### 4.2. General Description

An Intelligent System for Sound Annotation (ISSA) solution was developed to support the labeling of audible daily activities at home. ISSA works standalone and continuously listens to sounds and gets feedback from the user to annotate audible activities in real-time. ISSA is integrated with elements such as a sound detector, an audio classifier and voice assistant functionalities.

#### 4.2.1. ISSA: A Conversational Agent for Data Annotation

ISSA has been deployed in a Raspberry PI 3 B+ board and a MATRIX Voice, a seven-microphones array with integrated Analog Digital Converter (ADC), which allows capturing of high-quality audio in a digital form. We used the MATRIX HAL (Hardware Abstraction Layer) library to configure the features (sampling frequency, the minimum threshold for detection and duration) to detect and record sounds. Besides, we used ODAS (Open embeddeD Audition System), which implemented algorithms to get additional information about the sounds detected by the microphones such as 3D source direction estimation and level of energy. The audio classifier was implemented using Hidden Markov Models (HMM) and Mel Frequency Cepstral Coefficients (MFCC) as audio features, which are commonly used in literature for speech recognition and, as described in [34], work effectively in the proposed scenario of environmental sounds coming from a single source.

To build the voice assistant, we used cloud cognitive services such as API Speech and Dialogflow by Google and Text-To-Speech by IBM Watson. Using these tools, a voice assistant was built whose main task is to gather information from the user to label the audible activities detected in a specific environment. ISSA uses pre-trained models to classify and recognize sounds; however, for a future version, these classification models should be updated automatically using the data gathered to expand the range of audible activities recognized by ISSA.

Figure 6 shows an overview of the interaction between ISSA’s components, users and the environment. The description of how ISSA operates to annotate a specific audible activity is presented below. The detail of the parameters and threshold are detailed next.

The energy level for active and inactive sound sources follows a Gaussian distribution with means μActive=0.45 and μInactive=0.30. These parameters were set based on the values proposed by the ODAS library [36] and previous experiments. When ISSA detects a level of energy above the μA, it starts recording the sound. The following criteria were defined to stop the recording: Either the sound energy decreased below the μI or the sound was recorded for a maximum of 5.20 s. The duration of 5.20 s was determined from the average length of the audible activities considered. At this stage, additional information about the audio source—such as source direction, timestamp and duration—is obtained. ISSA ignores sounds whose energy does not overstep the μA.Once the sound is recorded, features and 3D spatial direction are extracted and sent to the Audio Classifier Component of ISSA. The classifier tries to recognize the recorded sound as an audible activity in real-time based on a previous training, detailed below:
The features are evaluated in all HMMs previously trained for each class and find the predicted class with the highest likelihood. One of the purposes of ISSA is adding new classes that are not previously trained. A way to know if an input sound class has not been trained is if its likelihood is low in all trained models. We defined two thresholds empirically: Prediction threshold $th_{p}$ and unknown class threshold $th_{uc}$. If the highest likelihood is higher than $th_{p}$, then the sound is classified as that class. If the likelihood is lower than $th_{uc}$, then the sound is treated as a new class which has not be trained. When the likelihood falls between both thresholds, ISSA will ask the user if the sound belongs to the class of the model with the highest likelihood.We compute the distance of the spatial direction of the incoming sound with the centroid spatial direction (previously computed) of the class with the highest likelihood. If the distance is larger than the centroid threshold $th_{c}$ (described below), the predicted class is rejected. If ISSA is in verbose mode, it queries the user, asking: What was that sound?If the distance between the spatial direction is smaller than the threshold $th_{c}$, we compare the likelihood with $th_{p}$. If the likelihood is equal or higher than the threshold, the system labels the sound as the predicted class.If the likelihood is not higher, but lies between $th_{uc}$ and $th_{p}$ and ISSA is in verbose mode, it asks the user to confirm the prediction with the question: Was that a <class X sound>?Finally, if the likelihood is lower than $th_{uc}$ and ISSA is in verbose mode, it queries the user asking: “What was that sound?”, at this point the user can also answer with: Which sound?, to which the system will respond by reproducing the sound that was recorded.Each time a new type of sound is identified, ISSA creates a new sound class with the given label.If ISSA is in silent mode, it does not ask anything to the user.ISSA interacts with users via the Voice Assistant. Using natural language processing, ISSA validates its inference or adds a new kind of audible activity using the confirmation or labeling mode, respectively. In addition, ISSA asks about the semantic position of the activity, such as the bathroom or kitchen.Finally, ISSA stores the instance of the audible activity. The complete audio data instance ($s_{i}$) has the following structure (Equation (Equation 1)):
(1)$$\begin{array}{c} s_{i}=\left\{\;activityName_{i},audioChannel_{i},3DPosition_{i},semanticPosition_{i},timestamp_{i}\right\}\; \end{array}$$
where activityName is the activity related to the sound, $audioChannel_{i}$ is an array with the seven sounds recorded by each microphone, $3DDirection_{i}$ is the spatial information according to the place where the sound was produced, semanticPosition is the location provided by the user where the sound was detected and $timestamp_{i}$ is the time when the sound was recorded.

### 4.3. Experimental Design

A description of the process of creating the Audio Classifier Component of ISSA is presented, including data collection and model training.

#### 4.3.1. Data Collection

The initial version of the Audio Classifier is limited to the one-person/one-sound approach, i.e., the assumption is that only one subject lives in the home. Audio produced by an individual was collected in a flat (see Figure 7), in which the microphones array was placed at a central location concerning the layout of the scenario. This location was considered as the place in which most audio produced at the flat could be detected. Several threshold (active, inactive) configurations to detect sounds were tested before gathering data to select the most appropriate ones for the scenario.

Table 1 shows the sounds that were annotated and the locations in which they were produced. The selection criteria to consider these sounds was that they are typically associated to activities of daily living that have audible manifestations.

We collected ten samples from the eight different home activities resulting in a dataset of 80 samples, each of them with duration ranging between 2.5 and 5.2 s. We additionally collected five different recordings with unsegmented audio where each recording included an instance of all classes. The dataset and its detailed information have been published in https://github.com/dagocruz/audible-home-activities-dataset.

#### 4.3.2. Model Training

As defined in Equation (Equation 1), a data instance sample, for example for the “Flushing toilet” sound, is represented as:(2)$$\begin{array}{c} s_{i}=\left\{\;flushingToilet,audioData,\left[0.61,0.49,0.60\right],2019-04-25T08:51:31.273Z\right\}\; \end{array}$$

We collected the 3D directions from sounds coming from fixed locations (blender, sink, toilet, shower, door, bathroom sink). Figure 8 shows their directions with respect to the space in the proposed scenario. Centroids of spatial direction were computed by averaging the samples for each class. The maximum distance between the spatial direction of a sound and its corresponding centroid was 0.1798 (the case of the blender). A 0.2 radius threshold for centroids was established.

As it can be noted, it is possible to only have centroids for the sound sources with fixed locations. In the case of the moving sound sources, such as walking or brushing teeth, [0,0,0] is indicated as a coordinate and a very high threshold (10,000). Therefore, the sounds likely to belong to a class with a moving sound source will be evaluated only by their likelihood.

To train the HMMs, the channel eight delivered by the device was used, that is, a single signal after applying beamforming processing with the seven recording channels. The audio collected and annotated was recorded at a sampling frequency of 16,000 Hz. The analysis of the 80 samples of annotated data was carried out by extracting the audio features MFCC using the Python-speech-features library with the default library parameters (window length = 25 ms, step = 10 ms, number of Cepstrum = 13, FFT size = 512). Additionally, we trained Gaussians HMMs (ergodic, four states) for each class using the Python HMM-learn library.

### 4.4. Results

One HMM was trained for each of the eight classes. To classify a sound, its likelihood score from each HMM is estimated and it is classified as the label of the HMM with the highest estimated likelihood. A three-cross-fold validation was performed with the training data resulting in a 96.20% accuracy value. In the classification, a sink sound was confused with a shower sound and a door sound was confused with walking twice. In those errors, the spatial direction is useful to identify the misclassification because in both cases the directions are well separated.

The setting of both thresholds, $th_{p}$ and $th_{uc}$, brings a trade-off in the types of errors in the classification. A low prediction threshold, $th_{p}$, will allow some incorrect classifications, but will never miss a sound classified as its true class. A high unknown class threshold, $th_{uc}$, will identify many of the sounds as not trained, so ISSA would ask the user many times about the class of the sound, but would reduce the number of incorrect classifications. For this reason, the thresholds should be adjusted depending on the accepted trade-off according to the user preferences.

We conducted an evaluation of the proposed approach at the same flat where the training dataset was obtained, and the user was instructed to perform one single activity at a time. Thus, the user generated five different continuous sequences of sounds, including those of the classes used for training, while ISSA was working in real-time to detect, classify and annotate each activity. To evaluate ISSA’s performance against unknown sounds, three unclassified sounds were included: “ringing telephone”, “typing on a keyboard” and “teapot whistling”—denoted as the “unknown sounds” class. Two conditions were compared for ISSA. First, ISSA classified sounds only using features from the sound; and a second condition where ISSA added the 3D direction estimation to perform the classification.

For the first condition, ISSA achieved an accuracy of 51.90%. While in the second condition, where ISSA used the 3D direction estimation of the sound source, accuracy increased to 87.27% of accuracy (see Figure 9). Thus, there is evidence of the impact of the use of the 3D direction of the source to get a better classification for an incoming sound. However, there are issues related to those sounds (e.g., sink bathroom and shower) which have similarities both in sound features and in 3D direction estimation.

### 4.5. Discussion

In this section, we describe an approach for the semi-automatic labeling of environmental audio data. We use a device with microphones, processing capabilities and speakers to interact with the user/annotator. The speaker is used to query the user, who labels the sound registered by the system by voice. We performed an initial evaluation to test the feasibility of the system in recognizing current trained classed and adding new sound classes. As we can see, the current deployment still confuses flushing sounds with shower running water and the unknown sounds with washing teeth and walking steps. In the case of the water sounds, both have a similar location and their sound is alike so it is hard for the system to identify the difference. A possible solution would be the use of another array of microphones in a more suitable location to distinguish them, or the generation of better models with a larger dataset. On the other hand, the other sounds have no fixed locations, so their direction information is not useful to improve classification. In these cases, larger valuable databases and better models will improve classification results.

## 5. Conclusions

Semi-automatic labeling provides a low-burden approach to obtain ground truth to train supervised classification systems to infer activity, behavior and mood for pervasive healthcare systems. These approaches allow recognition systems to be trained with an initial set of labeled samples, and to gradually label additional data to improve classification accuracy.

We described two approaches for self, online, naturalistic labeling and present results of experiments to assess their feasibility. While both studies have limitations (fixed arm position in the gesture recognition study and one single activity at a time in a single apartment in the second study), they achieve 80% to 90% precision. In addition, both approaches include strategies to correct labeling mistakes. This makes them feasible solutions to reduce the burden of large-scale data collection and labeling, in which some labeling errors are expected. The second approach is particularly appropriate for building incrementally more robust classifiers, by requesting the user to correctly identify new samples of classes that have exhibited low precision or are underrepresented in the dataset.

However, just as individual studies have their limitations, in general the semi-automatic annotation approach suggests a number of drawbacks. The main issue of the semi-automatic approach is to determine when to request feedback from the participants, which is currently done with thresholds. For example, if the ISSA system, of the second study, is used in the home of a new user, many of the sounds of this new environment may not be accurately recognized (even for sounds with a trained class, e.g., flushing toilet), so additional effort from the user to label the new instances is required initially. However, definitely, during this retraining, the system should not request the labeling of any audible sound; therefore, it is necessary to determine when it is convenient to request the intervention of the participants.

On the other hand, although the system of the second study is designed to operate continuously, the environmental noise can generate a large number of instances of no interest; in addition, some classes have high intra-variability, such as music or television audio.

## 6. Future Directions

Semi-automatic labeling requires both the participation of users and an automatic entity that queries the user based on an initial inference of an event. Future work is required in the users’ trench to define the proper query strategies to avoid user burden. For this, Human–Computer Interaction studies are necessary to evaluate how questions should be asked and how responses should be collected. For instance, the two studies presented in this paper could be merged to improve a semi-automatic labeling system by allowing yes/no responses to ISSA with predefined hand gestures. On the other hand, future work is also required to improve the automatic component. This can be achieved by using and developing better algorithms for the initial guessing of the events labels. Deep learning has been recently succeeding in different fields, including sound classification, so exploring the use of these algorithms in ISSA to improve precision is a worthy pursuit. However, a larger dataset is required, thus our proposed semi-automatic labeling contributes to this matter. Future directions also include defining strategies for re-training automatic algorithms. For instance, when a class has reached N new labeled instances, a new model should be generated. Moreover, the cocktail party problem is a current problem to be assessed in the field of environmental sound recognition. An advantage of ISSA is that it uses a microphone array that allows the recording of sounds from different locations. In any case, the advances in this topic should be integrated in the system.

An interesting possibility of ISSA in the domain of pervasive healthcare is the ability to map the semantic location of sounds, such as bathroom and kitchen. For instance, it would allow knowing if a person changes their behavior by changing the semantic locations where they perform a given activity.

## Figures and Tables

**Figure 1 sensors-19-03035-f001:**
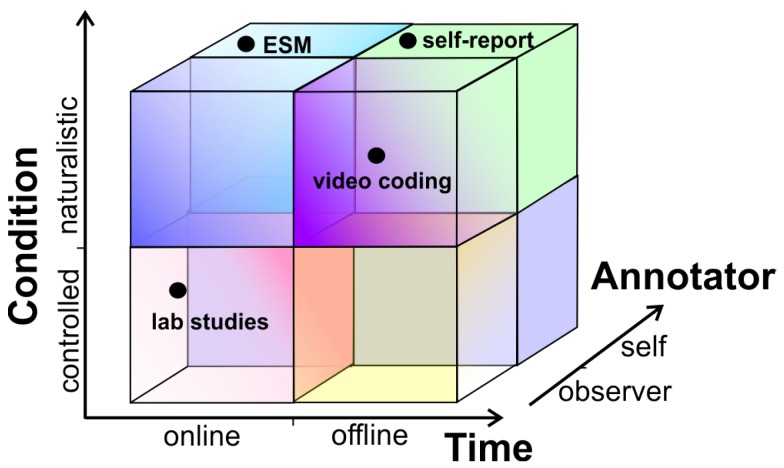
Annotation approaches classified in three dimensions: Time, author of the annotation and ecological validity.

**Figure 2 sensors-19-03035-f002:**
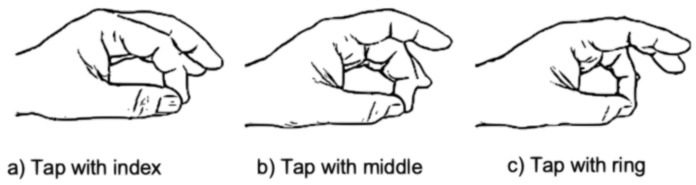
Three of the subtle gestures selected for data labeling. They require users to tap their thumb once with either the index, middle or ring finger. The three additional gestures are similar but require a double tap from the user.

**Figure 3 sensors-19-03035-f003:**
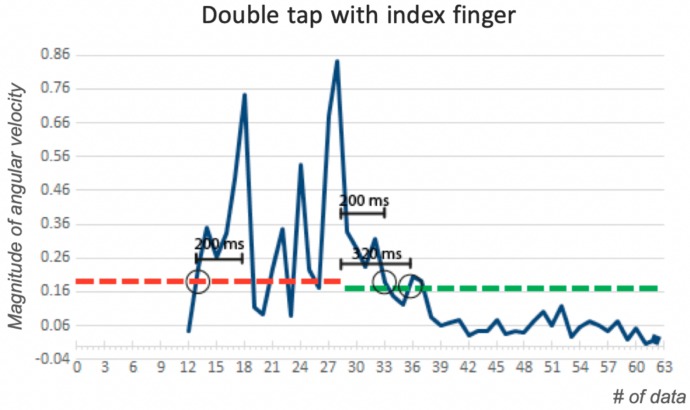
Estimating the end of the gesture. A peak is detected and eliminated at the end of the signal to avoid movements not associated with the gesture. The final signal, of 800 ms, goes from data point 13 to 33.

**Figure 4 sensors-19-03035-f004:**
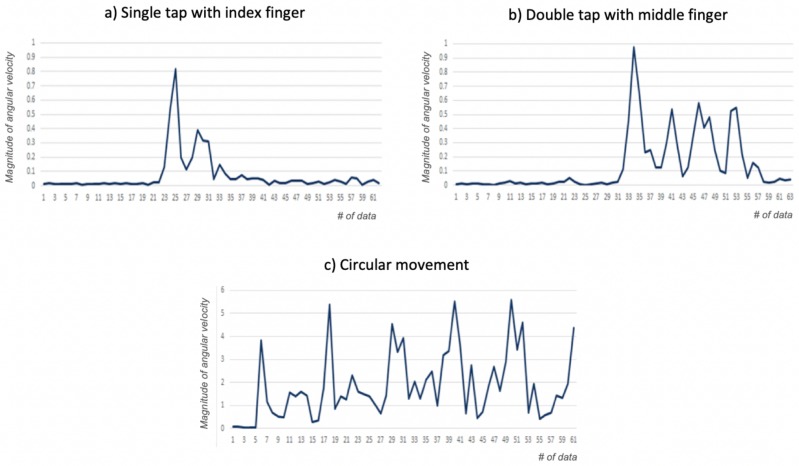
Recognizing the presence of a gesture using DTW. (**a**) Signal of the magnitude of angular velocity used as a reference (single-tap with index finger). (**b**) Signal in which the presence of a gesture is correctly inferred (double-tap with ring finger). (**c**) Signal from a hand performing another movement (circular movement with the wrist).

**Figure 5 sensors-19-03035-f005:**
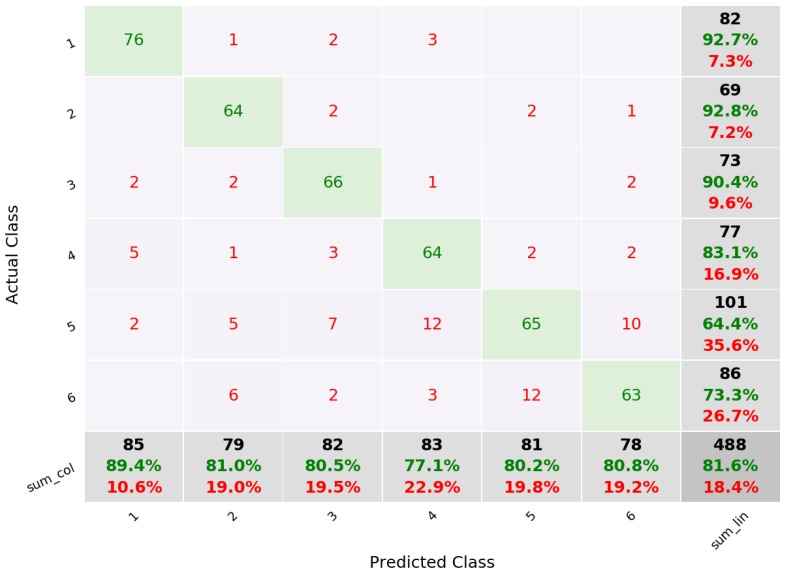
Confusion matrix for 6 gestures using SVM. 1—index, 2—middle, 3—ring, 4—double index, 5—double middle, 6—double ring.

**Figure 6 sensors-19-03035-f006:**
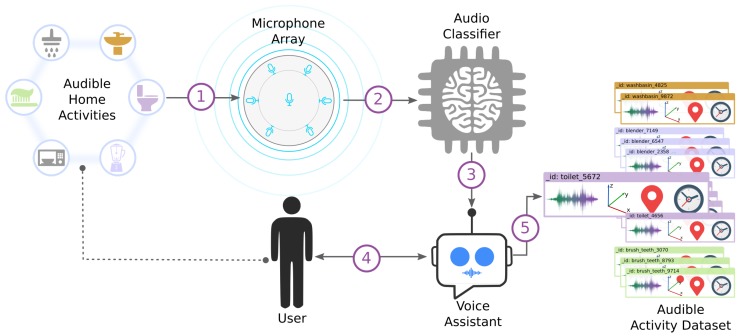
Overview of the functionality of Intelligent System for Sound Annotation (ISSA).

**Figure 7 sensors-19-03035-f007:**
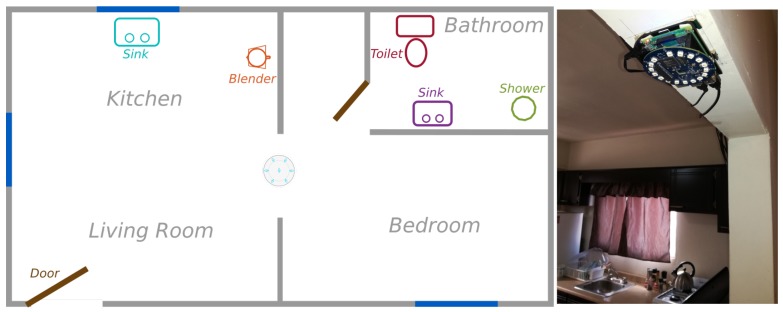
Setup used to record audio samples of home activities in the study.

**Figure 8 sensors-19-03035-f008:**
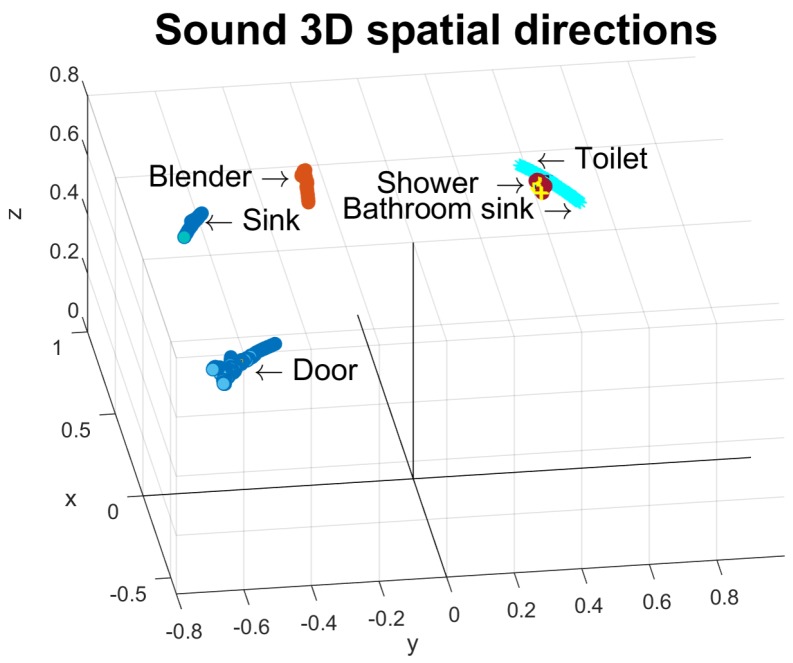
Spatial 3D direction from collected sound samples.

**Figure 9 sensors-19-03035-f009:**
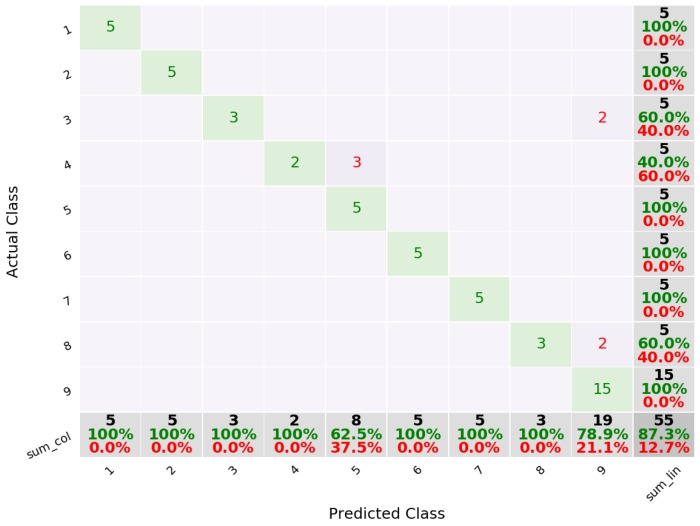
Confusion matrix for 5 continuous sequences using sound features and 3D direction estimation. 1—Running water from sink, 2—blender working, 3—walking steps, 4—running water from bath sink, 5—running water from shower, 6—flushing toilet, 7—door, 8—brushing teeth, 9—unknown sounds (ringing telephone, typing on a keyboard and teapot whistling).

**Table 1 sensors-19-03035-t001:** Audible home activities annotated and the locations in which they were produced. Living room: LIV; kitchen: KIT; bathroom: BATH; anywhere: ANY.

Id	Sound	Location
1	Running water from sink	BATH
2	Blender working	KIT
3	Walking steps	ANY
4	Running water from bath sink	BATH
5	Running water from shower	BATH
6	Flushing toilet	BATH
7	Door	LIV
8	Brushing teeth	ANY

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
