# Peer review of "Semi-Automated Data Labeling for Activity Recognition in Pervasive Healthcare"

_sensors, 2019, doi:10.3390/s19143035_

Round 1
Reviewer 1 Report
The paper presents model of health monitoring by the use of machine learning approaches. In general the paper is interesting so after changes possibly accepted for publication.
- Abstract is wide and does not present the research model well, please rewrite it.
- Introduction should be revised for most recent advances in the field general support in smart environments like Bio-inspired voice evaluation mechanism, A smartphone application for automated decision support in cognitive task based evaluation of central nervous system motor disorders, A Multi-Agent Gamification System for Managing Smart Homes.
- Present your SVM model in details: what kind of function have you used, what kind of support space have you chosen, etc.
- The data used for your experiment is not described well, discuss you data sets.
- It is not clear for me which kind of gestures have you used in your experiments? These are only those 3 gestures presented in figures?
- On the other hand you present 10 features in your confusion matrix, so which is the exact number of features in your paper?
- I don’t understand your fig 7. Does it present a scenario ?? For me it is an equipment used to capture the gesture. Revise caption of this figure.
- You present sound processing, so how do you mange capturing the sounds and transforming them to digital form used in the system.
- Eq (1) is different from eq (2), does it mean you change the number of features during processing?
Author Response
The paper presents model of health monitoring by the use of machine learning approaches. In general the paper is interesting so after changes possibly accepted for publication. Point 1: - Abstract is wide and does not present the research model well, please rewrite it. Response 1: The abstract of the manuscript was rewritten highlighting the research approaches of the study: “Activity recognition, a key component in pervasive healthcare monitoring, relies on supervised classification algorithms that require labeled data of individuals performing the activity of interest to train accurate models. Labelling data can be performed in a lab setting where an individual enacts the activity under controlled conditions. The ubiquity of mobile and wearable sensors allows the collection of large datasets from individuals performing activities in naturalistic conditions, where accurate labels are difficult to obtain. Gathering accurate data labels for activity recognition is typically an expensive and time-consuming process. In this paper we present two novel approaches for semi-automated online data labeling performed by the individual executing the activity of interest. The approaches have been designed to address two of the limitations of self-annotation: (i) the burden on the user performing and annotating the activity, and (ii) the lack of accuracy due to the user labeling the data minutes or hours after the completion of an activity. The first approach is based on the recognition of subtle finger gestures performed in response to a data labeling query. The second approach focuses on labeling activities that have an auditory manifestation and uses a classifier to have an initial estimation of the activity, and a conversational agent to ask the participant for clarification or for additional data. Both approaches are described, results from controlled experiments to assess their feasibility are presented, and their advantages and limitations are discussed.” Point 2: - Introduction should be revised for most recent advances in the field general support in smart environments like Bio-inspired voice evaluation mechanism, A smartphone application for automated decision support in cognitive task based evaluation of central nervous system motor disorders, A Multi-Agent Gamification System for Managing Smart Homes. Response 2: Recent studies have been included in the introduction as possible applications to which the current work can be focused, including those suggested by reviewer 1: “... ubiquitous computing systems increasingly rely on the inference of the activity and behavior of individuals to proactively assist users in their lifestyle [2], to continuously monitor the health of patients [3], or even to estimate symptoms of certain diseases [4]”. Point 3: - Present your SVM model in details: what kind of function have you used, what kind of support space have you chosen, etc. Response 3: As indicated in Section 3.2.2 we used the two most commonly used kernels for SVM, linear and radial basis functions. We trained the classifier with several parameters for cost and gamma, following the approach proposed by [Hsu, 23], That is, we varied cost and gamma exponentially (C={2-5, 2-4, 2-3, … 215}; gamma = {0, 2-15, 2-14, 26}). We identified between which values the best results were obtained and further subdivided these parameters to find the parameters that provided the best results, which corresponded to a cost of C=4 and g = 0.0078125. Point 4: - The data used for your experiment is not described well, discuss you data sets. Response 4: We included a new section 3.3.1 named “Data collection” to summarize the dataset for the first study: The dataset of the first experiment consists of a total of 540 samples. These included two instances of each of the 6 gestures performed by each of the 15 participants in three different postures (2x6x15x3 = 540). The dataset for the second experiment includes 100 samples, these are the gestures recorded by each of the 5 participants answering the 20 questions asked. Each sample in each dataset includes 15 signals, corresponding to the X,Y, and Z axis of the accelerometer and gyroscope recording, in addition to the ground-truth corresponding to the gesture performed. We added a summarization of the collected data information in the second experiment at the end of the section 4.3.1 to clarify the description of the used data: We collected 10 samples from the 8 different home activities resulting in a dataset of 80 samples, each of them with duration ranging between 2.5 to 5.2 seconds. We additionally collected 5 different recordings with unsegmented audio where each recording included an instance of all classes. The dataset and its detailed information have been published in https://github.com/dagocruz/audible-home-activities-dataset. Point 5: - It is not clear for me which kind of gestures have you used in your experiments? These are only those 3 gestures presented in figures? Response 5: We have modified the caption in Figure 2 and the corresponding text in the paragraph above to clarify that we are showing three of the gestures, and the additional three gestures are similar, but tapping twice with the thump, rather than just once. Point 6: - On the other hand you present 10 features in your confusion matrix, so which is the exact number of features in your paper? Response 6: The number of features for the first study is 208 as indicated in section 3.4.2. We have modified the text of the caption in Figure 5 (confusion matrix) to clarify things. The confusion matrix shows how the accuracy of the classification of the 488 samples recorded. Cross validation was used as resampling procedure for the evaluation with a k=10. Point 7: - I don’t understand your fig 7. Does it present a scenario ?? For me it is an equipment used to capture the gesture. Revise caption of this figure. Response 7: Thank you for the observation. We have corrected the caption to clarify that this is not a Scenario, but the experimental setup to record audio samples of home activities in the study. Point 8: - You present sound processing, so how do you mange capturing the sounds and transforming them to digital form used in the system. Response 8: We modified section 4.2.1 as following: ISSA has been deployed in a Raspberry PI 3 B+ board and a MATRIX Voice, a 7-microphones array with integrated Analog Digital Converter (ADC) which allows capturing high-quality audio in a digital form. We have also added the following in section 4.2.1: We used MATRIX HAL (Hardware Abstraction Layer) library to configure the features (sampling frequency, the minimum threshold for detection, and duration) to detect and record sounds. As we describe in section 4.3.2, we use 16Khz of sampling frequency to sample audio signal. Point 9 :- Eq (1) is different from eq (2), does it mean you change the number of features during processing? Response 9: Equation (1) shows how is the archetype of audible instances describing its structure. Equation (2) is an example of a specific instance of a flushing toilet activity, activityName in this case is FlushingToilet, 3DPosition is [(0.61, 0.49, 0.60] and so on.
2. Winnicka, A.; Kesik, K.; Polap, D.; Wozniak, M.; Marszałek, Z. A Multi-Agent Gamification System for Managing Smart Homes. Sensors 2019,19.
3. Bravo, J.; Hervás, R.; Fontecha, J.; González, I. m-Health: Lessons Learned by m-Experiences.Sensors 2018, 18.
4. Lauraitis, A.; Maskeliunas, R.; Damasevicius, R.; Polap, D.; Wozniak, M. A smartphone application for automated decision support in cognitive task based evaluation of central nervous system motor disorders. IEEE Journal of Biomedical and Health Informatics 2019, pp. 1–1.
Reviewer 2 Report
The paper presents a data labeling strategy for pervasive healthcare. The paper is an extension of a previously published paper and presents new data in the evaluation of the prototypes.
- In introduction, I suggest to include a last paragraph explaining the structure of the paper to the reader (what every section presents in a short form)
- The introduction has only 2 references, this way, I suggest the authors to include a better introduction of the problem (between lines 39 and 40) with recent related research
- The same problem is presented in sections 2.1 and 2.2. Please insert examples and research that demonstrate the approaches to data labeling as mentioned.
Author Response
The paper presents a data labeling strategy for pervasive healthcare. The paper is an extension of a previously published paper and presents new data in the evaluation of the prototypes. Point 1: - In introduction, I suggest to include a last paragraph explaining the structure of the paper to the reader (what every section presents in a short form) Response 1: A new paragraph has been included at the end of the Introduction section: The remainder of the paper is organized as follows: Section 2 presents the related work in terms of approaches to data labeling and considering different types of criteria (temporal, annotator, scenario and annotation mechanisms). Section 3 describes and presents the results of the first labeling study presented, which is about the use of gesture recognition with smartwatches. Section 4 describes and presents the results of the second labeling study presented, which is about the use of smart microphones to label audible home activities. Finally, conclusions and future directions are presented in Sections 5 and 6, respectively. Point 2: - The introduction has only 2 references, this way, I suggest the authors to include a better introduction of the problem (between lines 39 and 40) with recent related research Response 2: We have modified the introduction to more clearly introduce the problem and included additional references. The referred paragraph has been modified as follows: Data labeling is a demanding task that is normally performed either through direct observation of the individual performing the activity or via self-report. While the former approach is costly and impractical for large populations, the latter is error-prone [6]. Data labeling using direct observation is often performed in a lab, where individuals enact an activity, such as walking or sleeping, under controlled conditions. These conditions can affect how the activity is performed and the approach also has low ecological validity [7]. The use of self-report is convenient to label datasets from large numbers of individuals performing the activity in naturalistic conditions. The limitations of self-report however, have been well documented and include poor recollection and social-desirability biases. This problem is exemplified by recent criticism on the use of self-report dietary data in nutrition research and public policy [8]. The ubiquity of mobile and wearable sensors is allowing researchers to collect large datasets from individuals performing activities in naturalistic conditions to train activity classification models. Labeling this data however is particularly challenging and expensive. It requires self-labeling approaches that reduce the burden on the user and response bias. In addition, section 2 has been expanded to better explain the state of the art in data labeling and motivate the problem addressed. 6. Bulling, A.; Ward, J.A.;Gellersen, H. Multimodal recognition of reading activity in transit using body-worn sensors. ACM Transactions on Applied Perception, 2012, 9, 2. 7. Schmuckler, M.A. What is Ecological Validity? A Dimensional Analysis. Infancy, 2001, 2, 419-436. 8. Mitka, M. Do Flawed Data on Caloric Intake from NHANES Present Problems for Researchers and Policy Makers? Journal of the American Medical Association, 2013, 310, 2137-2138. Point 3: - The same problem is presented in sections 2.1 and 2.2. Please insert examples and research that demonstrate the approaches to data labeling as mentioned. Response 3: o The following text was added in the “Temporal (when: online vs offline)” subsection in the paragraph starting with the sentence: “Data can be labeled when the activity of interest…”: § The approaches described in [11] and [12] are examples of this criteria. § The approaches presented in [13], [14] and [15] describe tools for offline data annotation. o The following paragraph was amended to add related work: § A recent study found that self-annotation may not be an effective way to capture accurate start and finish times for activities, or location associated with activity information [16]. Although in this study participants were asked to proactively label their activities, not in response to a query. In [12], users manually label the activities they perform in their normal routine using a mobile app. On the other hand, recruiting people to annotate large amounts of data can be costly, or the observer could not have enough context to accurately label the data. The data labeling approach presented in [17] enables crowds of non-expert workers, volunteer and paid, to assign labels to activities performed by users in a controlled environment. 11. Schroder, M.; Yordanova, K.; Bader, S.; Kirste, T. Tool support for the online annotation of sensor data. Proceedings of the 3rd International Workshop on Sensor-based Activity Recognition and Interaction. ACM, 2016, p. 9. 12. Cruciani, F.; Cleland, I.; Nugent, C.; McCullagh, P.; Synnes, K.; Hallberg, J. Automatic Annotation for Human Activity Recognition in Free Living Using a Smartphone. Sensors, 2018, 18, 2203. 13. Kipp, M. ANVIL - a generic annotation tool for multimodal dialogue. Proceedings of the 7th European Conference on Speech Communication and Technology, 2001, pp. 1367-1370. 14. Cowie, R.; Sawey, M.; Doherty, C.; Jaimovich, J.; Fyans, C.; Stapleton, P. Gtrace: General trace program compatible with emotionml. Proceedings of the Humaine Association Conference on Affective Computing and Intelligent Interaction. IEEE, 2013, pp. 709-710. 15. Brugman, H.; Russel, A. Annotating Multi-media/Multi-modal Resources with ELAN. Proceedings of the 4th International Conference on LAnguage Resources and Language Evaluation; European Language Resources Association (ELRA): Lisbon, Portugal, 2004; pp. 2065-2068. 16. Tonkin, E.; Burrows, A.; Woznowski, P.; Laskowski, P.; Yordanova, K.; Twomey, N.; Craddock, I. Talk, Text, Tag? Understanding Self-Annotation of Smart Home Data from a User’s Perspective. Sensors, 2018, 18, 2365. 17. Lasecki, W.S.; Song, Y.C.; Kautz, H.; Bigham, J.P. Real-time crowd labeling for deployable activity recognition. Proceedings of the 2013 conference on Computer supported cooperative work. ACM, 2013, pp.1203-1212.
Reviewer 3 Report
In activity and behaviour recognition using classification, especially in pervasive healthcare systems, it is important to train classification algorithms with the ground truth, i.e., labelled dataset. The ground truth used in activity recognition has largely been manually labelled. And according to the authors, the process of manually gathering accurate labelled dataset is very expensive and time consuming. In order to address these problems and also that of the burden on the user performing and annotating the activity data and the lack of accuracy due to the user being asked to label the data minutes or hours after the activity has been completed, the authors proposed two methods for semi-automating the online labelling of activity recognition data. The first method uses subtle finger gesturing as a means to responding to data labelling query, whereas the second approach focuses on activities that have an auditory manifestation and uses a classifier to make an initial estimation of the activity and a conversational agent that asks the participant for clarification or to provide additional data.
I have the following comments about the strengths and weaknesses of the article.
1. The problem of obtaining the ground truth for activity and behaviour recognition is an important one, which has not been adequately addressed by the existing body of work. This article has proposed two methods as contributions to addressing this problem, not only to improve recognition accuracy but also to address other problems associated with manually labelling activity datasets for classification.
2. The article is well written and structured. The two approaches that have been proposed are well articulated and supported with experimental evaluations.
3. Of more interesting point is the categorisation of activity and behaviour data labelling based on time, annotation, annotation mechanisms, and annotation scenarios as described and presented by the authors.
4. Apart from using gesture recognition, using sound classification for data labelling is another important point of the article.
However, I have the following questions/concerns regarding the article that need to be addressed by the authors.
1. One of the weaknesses is that this article does not provide a comprehensive state-of-the-art. Providing different ways by which data can be labelled is not enough. Even claims by the authors in one of the categories (temporal data labelling) has not been supported with literature.
2. What informed the selection of support vector machine (SMO & SVM)? One would expect that having used WEKA for the experiment, the authors could have evaluated several machine learning algorithms.
3. Handcrafted feature extraction as used by the authors comes with some challenges, it might lead to poor classification performance. Why have the authors not considered automatic feature extraction using Deep Neural Networks such as Convolutional Neural Networks?
4. In sound classification for data labelling, also, only one algorithm (HMM) has been evaluated. One wonders why one algorithm? Experimentation with other algorithms would allow comparisons of their performances to inform which of them performs best with the data.
5. Minor typos such as “recorgint” on line 213 and “feasibilit” on line 271.
Author Response
In activity and behaviour recognition using classification, especially in pervasive healthcare systems, it is important to train classification algorithms with the ground truth, i.e., labelled dataset. The ground truth used in activity recognition has largely been manually labelled. And according to the authors, the process of manually gathering accurate labelled dataset is very expensive and time consuming. In order to address these problems and also that of the burden on the user performing and annotating the activity data and the lack of accuracy due to the user being asked to label the data minutes or hours after the activity has been completed, the authors proposed two methods for semi-automating the online labelling of activity recognition data. The first method uses subtle finger gesturing as a means to responding to data labelling query, whereas the second approach focuses on activities that have an auditory manifestation and uses a classifier to make an initial estimation of the activity and a conversational agent that asks the participant for clarification or to provide additional data. I have the following comments about the strengths and weaknesses of the article. 1. The problem of obtaining the ground truth for activity and behaviour recognition is an important one, which has not been adequately addressed by the existing body of work. This article has proposed two methods as contributions to addressing this problem, not only to improve recognition accuracy but also to address other problems associated with manually labelling activity datasets for classification. 2. The article is well written and structured. The two approaches that have been proposed are well articulated and supported with experimental evaluations. 3. Of more interesting point is the categorisation of activity and behaviour data labelling based on time, annotation, annotation mechanisms, and annotation scenarios as described and presented by the authors. 4. Apart from using gesture recognition, using sound classification for data labelling is another important point of the article. However, I have the following questions/concerns regarding the article that need to be addressed by the authors. Point 1: One of the weaknesses is that this article does not provide a comprehensive state-of-the-art. Providing different ways by which data can be labelled is not enough. Even claims by the authors in one of the categories (temporal data labelling) has not been supported with literature. Response 1: Thank you for your comments. This observation was pointed out and attended in points 2 and 3 by Reviewer 2; therefore, the references to related work were incorporated to Sections 1 and 2, in which the problem in the labeling of data is described and classified: Point 2: What informed the selection of support vector machine (SMO & SVM)? One would expect that having used WEKA for the experiment, the authors could have evaluated several machine learning algorithms. Response 2: We decided to use SVM since its one of the most widely used supervised classifiers for shallow machine learning. At an initial phase of the study we also experimented with backpropagation neural networks, but decided to stop using them since the training time was too large. We decided to use SMO, a variant of SVM, due to its efficiency and the feasibility of running the algorithm in the smartwatch. We agree that comparing several classifiers was feasible with WEKA, but our focus was on assessing the feasibility of the approach proposed, rather than obtaining the best classifier for our dataset. Point 3: Handcrafted feature extraction as used by the authors comes with some challenges, it might lead to poor classification performance. Why have the authors not considered automatic feature extraction using Deep Neural Networks such as Convolutional Neural Networks? Response 3: We agree that Deep Neural Networks have been currently producing good results in several fields and are becoming the state of the art in supervised classifiers. However, training this type of algorithm requires a large labeled dataset. In particular, deep learning suffers from over-fitting with small dataset [Pasupa, 2016]. As we mentioned in section 4.1, currently our dataset is relatively small and doesn’t come from heterogeneous scenarios. We used MFCC because it has been proven to work well in scenarios with “clean” data and a few classes (even close to a F1score of 100% [28]) as the proposed scenario. As we mentioned in the Future directions section: Deep learning has been recently succeeding in different fields, including sound classification, so exploring the use of these algorithms in ISSA to improve precision is a worthy pursuit. However, a larger dataset is required, thus our proposed semi-automatic labeling contributes to this matter. [Pasupa, 2016] K. Pasupa and W. Sunhem, "A comparison between shallow and deep architecture classifiers on small dataset," 2016 8th International Conference on Information Technology and Electrical Engineering (ICITEE), Yogyakarta, 2016, pp. 1-6. doi: 10.1109/ICITEED.2016.7863293 Point 4: In sound classification for data labelling, also, only one algorithm (HMM) has been evaluated. One wonders why one algorithm? Experimentation with other algorithms would allow comparisons of their performances to inform which of them performs best with the data. Response 4: We modified paragraph in 4.2.1 as following: The audio classifier was implemented using Hidden Markov Models (HMM), and Mel Frequency Cepstral Coefficients (MFCC) as audio features, which are commonly used in the literature for speech recognition and, as described in [28], work effectively in the proposed scenario of environmental sounds coming from a single source. In [28], we compared the performance of different features and different classifiers getting that MFCC-HMM works best with “clean” data as used in this scenario. We agree that comparison with other algorithms allows to encounter which performs best with the data. In this sense, a large data set helps to conduct experiments to achieve these comparisons. For that reason, our proposal of semi-automatic labeling serves as a means to collect data to ultimately encounter the most appropriate algorithm. Furthermore, inside ISSA, the classification stage is as a component that can use any classification algorithm. As we describe in the Conclusions, this allows to keep building incrementally more robust classifiers. As we mention in the Future directions section, there are still current problems to be assessed in the field of environmental sound recognition, such as the cocktail party problem. Point 5: Minor typos such as “recorgint” on line 213 and “feasibilit” on line 271. Response 5: The typos throughout the manuscript were carefully reviewed.
Round 2
Reviewer 1 Report
Authors revise a paper well so accept